# Determining the Relationship between Squat Jump Performance and Knee Angle in Female University Students

**DOI:** 10.3390/jfmk9010026

**Published:** 2024-01-29

**Authors:** Felice Di Domenico, Giovanni Esposito, Sara Aliberti, Francesca D’Elia, Tiziana D’Isanto

**Affiliations:** Department of Human, Philosophical and Education Sciences, University of Salerno, 84084 Fisciano, Italy; fdidomenico@unisa.it (F.D.D.); saaliberti@unisa.it (S.A.); fdelia@unisa.it (F.D.); tdisanto@unisa.it (T.D.)

**Keywords:** explosive strength, jump squat, kinematics, lower limb strength, knee angle measurement, motion analysis

## Abstract

The Squat Jump (SJ) test is widely recognized as a reliable test for assessing lower-limb explosive strength. However, uncertainty persists in the literature regarding the optimal starting positions for maximizing vertical jump performance. This uncertainty is exacerbated by a disproportionate focus on athletes in existing studies, with insufficient consideration being given to non-athletic women. To address this gap, this study investigated the influence of leg starting angle on explosive jump height in a homogeneous sample of non-athletic women. Thirty-two female students enrolled in a Sports Science master’s degree program at the University of Salerno participated in the study. Descriptive statistics were employed to summarize data on various variables, and Pearson’s correlations were calculated to assess the relationship between knee angle in the starting position and achieved jump height. The study revealed that different starting positions had a noteworthy impact on jump height among the participants. A strong negative correlation (−0.701) was identified between Squat Jump elevation and the knee angle in the starting position. Notably, 62.5% of the subjects opted for a starting knee position of approximately 70 degrees, with all of them consistently achieving a jump height associated with this specific angle. These findings provide valuable insights into the relationship between leg starting angle and explosive jump height in non-athletic women. The observed correlation underscores the significance of the starting position in Squat Jump performance. The prevalence of a specific knee angle choice among participants suggests potential implications for training and performance optimization in this sample.

## 1. Introduction

Jumping is a movement that requires coordination between the upper and lower body parts and is considered an indicator of functionality and general physical health that is useful in everyday life and performance in many sporting disciplines [1]. The propulsive action of the lower limbs during the thrust phase of the vertical jump employs the explosive force of the lower limbs [2,3,4]. Laboratory tests using instruments such as force plates, optoelectronic bars, and linear position transducers can provide insights into the physiological and biomechanical qualities of performance, and over the past two decades, laboratory tests have proven particularly useful for measuring, analyzing, and evaluating [5] the characteristics of explosive movements. The two tests used most often to measure the expression of lower limb power are characterized by vertical jump tests: the Squat Jump (SJ) test and the Countermovement Jump (CMJ) [6]. In the case of the former, the ability to express explosive force is measured, while in the case of the latter, the ability to express reactive elastic force is assessed [7,8,9]. Jumping tests provide important information regarding explosive and reactive strength and intermuscular coordination [10,11].

The various parameters recorded in jump performance measurements include jump height, measured in centimeters or inches, and relative peak power, measured in watt, as well as time of flight [12]. Jump height is an estimate of the change in height in the jumping individual’s center of mass, which is measured via the impulse–momentum equation [13,14] using special technological instrumentation. Measuring vertical jump height is useful and reliable for investigating power and explosive force in jumping [15], if the test conditions are met [16]. Although the Squat Jump (SJ) test is a simple, practical, valid, and reliable test for investigating the explosive strength of the lower limb explosive strength under predominantly concentric conditions [17], several factors that can significantly affect jump performance and alter the test measurement must be considered. Previous studies have reported different vertical jump performances at different starting angles and reported that jumps with very deep starts are not perfectly coordinated [18,19]. It is usually left to the individual to choose the depth of the jump start position. Several studies have shown, not always clearly and unanimously, the degree to which the starting position affects the final height achieved in the SJ test. Bobbert et al. [20] reported that a SJ performed from a shallower depth results in a greater jump height. Subsequently, Kirby and colleagues [21] also observed an increase in jump height performed at greater depths (approx. 75 cm) compared to shallower depths (15 cm). They hypothesized that this increase is due to a longer contraction time at greater depths, which allows athletes to apply force for longer periods, increasing the vertical impulse [13] (i.e., the product of applied force and time of application). These hypotheses, however, are not sufficiently supported by accepted scientific evidence. Recently, Gheller et al. [22] tested vertical jumps performed at different depths. This study’s sample consisted of thirteen male athletes, and SJs were performed with six different initial knee joint angles: 90°, 100°, 110°, 120°, 130°, and an angle of the athlete’s choice. The study showed that the best jumps in terms of elevation were those performed with a squat depth equal to 90° and those performed from an angle of the athlete’s choice. Jumping from a position with the knees bent at 90° seems to be the best starting position to adopt to achieve better performance [23].

The papers in the existing literature often provide conflicting data, as they often refer to sports subjects and differ both in terms of the parameters tested (bending depth or bending angle) and from a methodological point of view, as it is unclear whether or not the data acquisition procedures were influenced by instructions from the researchers and how much these procedures and the acquisition environment were protected from external perturbations. For example, Talpey et al. [24] compared two types of instructions given by the researcher to the test subject: one asked the subject to focus on jumping to reach their maximum height (instruction on jump height) and another instead asked the subject to focus on fast movement of the lower limbs to maximize explosive strength (instruction on leg tilt speed). Higher jump height and peak velocity values and a greater downward displacement were shown for the jump performed following the instruction that focused on height, while the instruction regarding focusing on leg speed provided higher maximum strength values. This shows that different instructions produce different responses in the execution of the test.

Furthermore, a necessary differentiation should be made between males and females when defining the ideal starting position for the SJ test. From puberty onwards, a different gender-related body conformation emerges [25] and correlates with hormonal influences that are substantiated by both the shape and position of the center of body mass (CoM) [26].

Thus, despite the number of studies in the literature, there is still much uncertainty as to which initial positions are best for achieving high levels of performance in explosive vertical jumping tests, as the results are conflicting. In addition, most studies have been conducted studies on male and female athletes who have exhibited a high-level mastery of the gesture as they have repeated and perfected the movement over many years, while non-athletic subjects have not been adequately considered. The evidence to date is not entirely clear, and the reason for this could be related to the non-homogeneity of the samples used and the use of different stimuli and feedback.

The aim of this study is to provide further evidence on the impact of the angle of the knee when in the starting position (and thus the cause-and-effect relationship between knee angle and SJ test jump height) on SJ test jump height by assessing a homogenous sample of female non-athletes with a standardized version of the test and without feedback from the researcher (i.e., letting the test subject choose the starting position independently). SJs appeared to be more appropriate than CMJs for achieving the study aim, as they allow for a more precise identification of the starting angle and its effects on the height of the vertical jump and, therefore, muscular explosiveness; other gestures, such as the CMJ, would have made it more difficult to standardize the acquisition procedure and would have provided data that would not have been useful for this study. The choice of a sample of female non-athletes was motivated by the need to undertake a preliminary study that would clarify the difference between female athletes and non-athletes, who have different physical abilities and skills in that the former, compared to non-athletes, tend to be more jump trained for sport-specific performance demands. Therefore, we aimed to identify the optimal angles for SJ execution in female subjects that can be applied in the training processes of female athletes who, while possessing optimal physical characteristics, do not have athletic experiences comparable to athletes of specific sports and therefore differ from athletes, who have been highlighted extensively in the scientific literature.

## 2. Materials and Methods

### 2.1. Study Design

This study was a preliminary study involving non-athlete female subjects with the intent to address the need to know the knee angles of subjects who differ from athletes in physical sport experience, training goals, and performance demands when in the starting position for the SJ test.

Ethical considerations were prioritized, and all participants were fully informed about the study’s objectives and procedures, with voluntary participation and informed consent being fundamental ethical pillars.

### 2.2. Study Participants

The participants in this study were carefully selected to ensure a homogeneous sample that could provide meaningful insights into the impact of starting leg position on Squat Jump (SJ) height among female non-athletes. Thirty-two female students, all enrolled in the Sports Science master’s degree program at the University of Salerno, participated in this research study. The participants, with an average age of 23.37 ± 0.94 years, average weight of 58.68 ± 1.92 kg, average height of 166.96 ± 2.07 cm, and average BMI of 21.05 ± 0.71, were chosen based on specific criteria to maintain uniformity within the sample.

The inclusion criteria included people with a minimum of two years of training experience who claimed to have had varying levels of experience in performing the act under study but who were not classifiable as athletes because they were not engaged in athletic competition but were engaging in physical activity for the benefit of their mental and physical well-being, ensuring a similar level of fitness without the potential biases introduced by specialized training. Additional inclusion criteria were established to maintain the integrity of the study, and these additional criteria included a body mass index (BMI) between 20 and 23, an age between 20 and 25 years, and the absence of health problems or injuries to the ankles, knees, hips, or back in the three months before the start of the study. The inclusion criteria aimed to eliminate confounding variables and create a focused study group. 

### 2.3. Instruments

The instrumentation employed in this study was meticulously selected to ensure the precision, reliability, and validity of the data collected. Each instrument played a crucial role in capturing specific aspects of the participants’ movements during the Squat Jump (SJ), contributing to a comprehensive understanding of the relationship between starting leg position and jump height.

The initial step in participant characterization involved the use of the Pegaso digital scale (GIMA Spa, Gessate, Italy). This instrument, renowned for its accuracy and reliability, provided anthropometric data essential for participant profiling. Height, weight, and other relevant measurements were recorded with precision, forming a foundational dataset for subsequent analyses.Then, the Optojump [27,28] optoelectronic device (Microgate Srl, Bolzano, Italy) was used in the data acquisition process, particularly for capturing the intricate dynamics of vertical jumping. Comprising a transmitter bar and a receiver bar, this state-of-the-art system offered real-time insights into various performance parameters. Positioned one meter apart, each bar, housing 96 LEDs with a remarkable resolution of 1.041 cm, facilitated precise data collection during the participants’ jumps. The Optojump system utilized optical sensing technology, allowing for the detection of interruptions in the infrared light between the transmitter and receiver bars. This interruption corresponded to the elevation of the participants during the jump, enabling accurate measurements of jump height. The choice to use the Optojump was driven by its ability to provide high-resolution, instantaneous data, essential for capturing the nuanced movements of the participants. To complement the optoelectronic data, a GoPro HERO7 camera [29] (GoPro, San Mateo, CA, USA) set to record at 120 frames per second (fps) was strategically incorporated into the experimental setup. This high-speed camera facilitated a detailed visual analysis of the participants’ movements during the Squat Jump test. The synchronized recording of the jump from multiple perspectives allowed for a comprehensive evaluation of form, technique, and the fluidity of motion. Placed three meters from the acquisition area and oriented perpendicular to the plane of action, the GoPro camera served as a reliable tool for capturing the fine details of the participants’ jumps. The footage was shot at 1440 p video resolution and 60 fps, a resolution that allowed for a linear field of view (FoV) (a setting built into the GoPro Hero 7 software version 1.90) in order to correct distortion errors. The chosen frame capture rate ensured that no subtleties of movement were overlooked, providing valuable insights into the biomechanics of the jumps.

#### 2.3.1. Kinovea Two-Dimensional Motion Analysis Software

The integration of Kinovea version 0.8, a sophisticated two-dimensional motion analysis software, elevated the post-acquisition phase of the study. This software played a pivotal role in the detailed examination of video recordings captured by the GoPro camera. Prior to analysis, a meticulous calibration procedure was followed, referencing the Optojump bar and fixing its value at 90 cm. Kinovea allowed for the precise measurement of kinematic parameters related to the knee angle at the starting phase of the Squat Jump. This involved marking three reference points—right lateral malleolus, right lateral condyle, and right greater trochanter—ensuring accuracy in angle measurement. The calibration process, coupled with the advanced features of Kinovea, facilitated a granular analysis of the participants’ movements, offering deeper insights into the influence of the starting leg position on jump height. Figure 1 shows the calibration process by referencing known measurements represented by the distance of the lines that demarcated the acquisition area. 

Figure 2 shows the digital marking process from the markers applied on the three landmark points and the subsequent calculation of the knee starting angle. This operation was performed for each participant.

The integration of anthropometric measurements, real-time optoelectronic data, high-speed visual recordings, and sophisticated motion analysis software ensured a multi-faceted approach to investigating the impact of the starting leg position on Squat Jump height. This comprehensive instrumentation strategy aimed not only to answer the immediate research questions but also to contribute valuable data to the broader scientific community interested in human movement analysis. The meticulous selection and integration of instruments reflected a commitment to scientific rigor and a nuanced understanding of the complexities involved in studying dynamic movements such as vertical jumping.

#### 2.3.2. Laboratory Setting

The study (Figure 3) was carried out in a suitably arranged controlled environment within a laboratory with dimensions of 6 × 4 × 4 m (length × width × height). Specialized PVC flooring was chosen to provide an optimal surface for the jumps, minimizing external variables that could affect performance. An acquisition area of 90 × 60 cm, within which the Optojump transmitter and receiver bars were strategically positioned, was demarcated on the laboratory floor. This spatial arrangement ensured that the participants’ movements were confined to a controlled area, contributing to the reproducibility of the tests.

The preparation of the laboratory included the careful positioning of the GoPro HERO7 camera on a stable tripod, ensuring a consistent focal point. The camera’s placement, three meters from the acquisition area and oriented perpendicular to the plane of action, facilitated optimal video recording for subsequent analysis. In addition, the laboratory setup incorporated a 10 min warm-up routine for participants, comprising joint mobility exercises and muscle stretching. This preparatory phase aimed to standardize the participants’ physiological states before the actual test, contributing to the reliability of the collected data.

### 2.4. Procedures

The procedural phase of this study was designed to ensure precision, reproducibility, and an environment conducive to capturing reliable data on the impact of starting leg position on Squat Jump (SJ) height among female non-athletes. Two days before the commencement of the study, participants received extensive briefings on the tests and procedures involved. This phase aimed to familiarize participants with our study expectations, mitigate anxiety, and ensure a consistent understanding of the execution technique. Participants were informed about the importance of their voluntary participation, the objectives of the study, and the significance of their role in contributing to scientific knowledge. On the day of the actual test, each participant underwent a test run to verify their comprehension of the execution technique. During this run, participants were instructed to jump as high as possible, ensuring a controlled landing in the starting position. This familiarization step aimed to enhance the accuracy and reliability of the subsequent jump trials. On the same day as the tests, participants’ biographical and anthropometric data were recorded, including age, height, weight, and body mass index (BMI). These data served as crucial descriptors of the study sample and were instrumental in subsequent analyses. The data acquisition, performed in June in a human movement analysis laboratory, ensured consistent testing conditions. The vertical jump height, a key parameter, was recorded using the Optojump system and the GoPro HERO7 camera. The participants, at the time of the test, were dressed in tight black elastic clothing. Three passive markers made of yellow tape measuring 1 × 1 cm were applied on three anatomical landmarks corresponding to the rotation axes of the hip, knee, and ankle. Knee angles in the early phase of the SJ were subsequently analyzed using Kinovea software. The markers were applied to all participants by the same kinesiologist, who is an expert in biomechanics with over 10 years of experience in motion analysis, and this kinesiologist also processed the images using Kinovea software with the identification of the rotation axes of the three marked joints. However, despite the robust experience of the kinesiologist, it is necessary to clarify that there may have been errors due to the movement of the clothing and, consequently, the movement of markers during the bending and straightening phase of the lower limbs. Such errors were minimized by repeating the test. Three trials of the SJ were executed for each participant within the defined acquisition area. The average result from these trials, carefully recorded in the dedicated software, formed the basis for subsequent statistical analyses.

#### Acquisition Protocol

The primary acquisition protocol focused exclusively on the Squat Jump (SJ). Participants were instructed to stand in the defined acquisition area, facing the short sides of the room and aligning their sagittal plane with the video acquisition plane. Before the jump, participants were in the starting squat position with their hands resting on the iliac crests, standardizing their upper body position. The starting position for performing the SJ required the position of the trunk to be parallel to the position of the ankles, observed in the sagittal plane. The starting position was maintained for 3 s before the test start signal. The Optojump program signaled the start of the jump through an audible signal. Participants performed a vertical jump upon hearing the signal, starting from a lower limb flexion position and keeping their hands fixed on their hips. At this stage, the participants maintained the autonomy to choose the degree at which they angled their knee while in the starting position, emphasizing a natural and uninhibited movement. Three trials were performed for each participant after a 120-s rest period, which ensured a robust dataset for subsequent analysis. The average result of these three trials was recorded, contributing to the complete dataset used for statistical analysis.

### 2.5. Statistical Analyses

The collected data were subjected to rigorous statistical analyses to derive meaningful insights into the relationship between the starting leg position and SJ height among female non-athletes. Descriptive statistics, presented as mean ± standard deviation, were employed to summarize the biographical and anthropometric data. Kinovea software facilitated the analysis of the video recordings, focusing on the measurement of knee angles at the starting phase of the SJ. Three reference points—left lateral malleolus, left lateral condyle, and left greater trochanter—were marked to ensure accuracy in angle measurement. Pearson correlation coefficients were then calculated to quantify the relationship between the knee angle in the starting position and the achieved jump height. These coefficients were classified as null (values from 0 to 0.2), weak (values from 0.2 to 0.4), moderate (values from 0.4 to 0.6), strong (values from 0, 6 to 0.8), and very strong (values from 0.8 to 1) [30]. Linear regression was used to evaluate the level of impact the starting knee angle had on the jump height; the variables considered were the starting knee angle (x) and the maximum height reached in the SJ test (y), and the adjusted R2 was used to measure the proportion of the change in the y variable compared to the change in the x variable. Statistical analyses were conducted using SPSS software (version 22; IBM, Armonk, NY, USA), with the significance level set at *p* ≤ 0.05. 

## 3. Results

The collection of biographical and anthropometric data aimed to establish a foundation for understanding the characteristics of the study participants. As seen in Table 1, the participants, with an average age of 23.37 years, exhibited a striking uniformity in their anthropometric attributes. The low standard deviations across age, weight, height, and BMI indicated a remarkably homogeneous sample, validating the efficacy of the inclusion and exclusion criteria in creating a cohesive study group. This homogeneity is crucial in ensuring that observed effects, such as those related to the starting leg position, are not confounded by significant variations in participants’ baseline characteristics.

Figure 4 shows the marking of reference points, carried out using Kinovea software, of individual frames acquired using a GoPro 7Hero camera. The starting angles of the SJ test were measured for each participant. The average angle recorded was 75.58 ± 8.32 degrees.

The focus of this study lies in the relationship between knee starting angles and jump height. The detailed measurements of knee angles at the starting phase of the Squat Jump (SJ) are presented in Table 2. 

The correlation coefficient of −0.701 revealed a high negative correlation between the knee angle in the starting position and the achieved jump height. This relationship was verified in 48% of cases in which for every centimeter less in the starting angle of the knee starting from 90 degrees up to 70 degrees, there was an improvement in the vertical jump of 0.25 cm. This significant finding suggested that variations in the starting position, particularly at the knee, had a substantial impact on the subsequent jump height among the study participants.

Figure 5 visually portrays the correlation between knee starting angle and jump height attained by female athletes in the Squat Jump (SJ) test. This graphical representation enhances the interpretability of the correlation coefficients, offering a clear depiction of the trends observed in the dataset. 

Observing the trendline provides additional context, illustrating the inverse relationship between knee flexion angle and jump height. The graphical representation enhances the accessibility of the findings, catering to both visual and quantitative interpretation. The categorization based on starting angles allows for a greater understanding of performance outcomes. Participants with starting angles less than or equal to 70 degrees recorded jumps of approximately 14 cm, while those with angles between 70 and 80 degrees achieved an average jump height of 16 cm. Notably, starting angles greater than 90 degrees resulted in lower performance, emphasizing the critical role of optimal knee flexion in maximizing jump height.

Figure 2 also visually identifies the optimal starting angle associated with the highest jump height. The graphical representation facilitates a direct observation of the knee flexion angle that yielded peak performance. In this study, the optimal starting angle was approximately 70.8 degrees, resulting in a jump height of 22.2 cm. This specific insight into the optimal starting position adds a practical dimension to the findings, potentially informing training programs and performance optimization strategies. Understanding the nuanced relationship between knee flexion angles and jump height enables targeted interventions aimed at enhancing explosive vertical jumping capabilities.

## 4. Discussion

The primary objective of this study was to explore the correlation between knee angle and jump height in Squat Jumps (SJs) and the cause–effect relationship between the two variables, specifically focusing on female university students who autonomously selected their initial positions. The study, conducted in a controlled laboratory environment, featured a homogeneous sample of non-athletes of a similar age who had similar anthropometric characteristics but characteristics that differed compared to those of athletes, who, due to sport-specific performance needs, are more familiar with the gesture examined. The selection of this sample was crucial for effectively elucidating the interdependence of variables, and the study implemented precise procedures with calibrated instruments.

The findings of the investigation demonstrated that the heights achieved in the jump test were significantly influenced by the diverse starting positions adopted by the participants. Notably, positions characterized by knee angles between 80 and 90 degrees and those below 70 degrees yielded suboptimal results. In contrast, starting angles at around 70 degrees produced superior jump heights. The study revealed a correlation index of -0.701 between Squat Jump elevation and knee angle in the starting position, indicating a strong negative correlation. This implies that as the starting squat angle increased, the jump height decreased. These results underscore the significance of squat depth, particularly at a knee angle of around 70 degrees, in shaping jumping performance. The results also showed that this model is applicable to 48% of cases, so for each degree of knee flexion, improvements of 0.25 cm were found.

However, it is important to acknowledge that the starting position is just one factor influencing vertical jump height. The entire gesture depends on the correct coordination of individual muscle groups [31,32], perfected through the learning process [33,34,35,36,37] in addition to other factors such as general strength, coordination, jumping technique, propulsion ability, as well as structural factors such as fiber arrangement, muscle transverse section, and the elastic and energy transduction properties of the subject’s various structures. The initial position of the SJ, at any hip and knee angle, involves the contraction of the lower limb muscles, which store elastic energy in the muscle fibers. From this angle of the knee and hip, the stored elastic energy is released during the bracing phase: the muscles of the lower limbs contract rapidly and extend the joints, converting the elastic energy into kinetic energy [38]. In the present study, the effects of only knee angle on jump height were examined, ignoring hip angles. Although hip angles are important [39], these data were not collected or addressed in this study, as we opted to simplify the image acquisition process and adapt it to the equipment in our possession. To minimize the influence of the hip on the execution of the SJ, before the test, the execution of the gesture was rehearsed by all participants under the guidance of expert kinesiologists to standardize the positions of the upper body. This release of elastic energy allows the body to jump upwards and reach jump heights dependent on the amount of energy transduced through this process. At greater depths, the time required to apply force to a point increases, generating a greater vertical impulse [21,40]. However, this effect on the vertical impulse was detected down to depths with starting knee angles of approximately 70 degrees. Probably, greater depths determine biomechanical limitations that are not conducive to optimizing the vertical jump. Pincivero et al. [41] found greater effects on absolute knee torque at angles between 70 and 90 degrees.

These factors differ between the two sexes from puberty onwards, where a higher proportion of fast muscle fibers (FT) is responsible for a greater amount of explosive force and greater leg length [42,43,44] (i.e., bony levers capable of producing greater moments of force have been shown to benefit males). Differences between the two sexes have been demonstrated, with females having less stiffness in some lower limb muscles than males [45] and therefore having a lower capacity to store and return elastic energy [46] during jumping. According to the results of this study, this effect in women is achieved more effectively with initial knee angles of approximately 70 degrees, and this is in line with other studies. For example, Brownstein et al. [47] identified the absolute peak of knee extensor torque during maximal-effort contractions at 50 degrees for young adult men and 70 degrees for young adult women. The data acquired in our study, collected using a GoPro HERO7 camera and processed using specific Kinovea software, showed angles varying from approximately 69 degrees up to approximately 95 degrees, indirectly proportional to the different jump heights detected using the Optojump Microgate optical detection system, which ranged from 12 to 22 cm height. A Squat Jump performed from a starting position with the knees at around 70 degrees probably requires greater activation of the lower limb muscles, particularly the quadriceps and calf muscles. O’Brien et al. [48] demonstrated peak joint moments at knee joint angles of 70 degrees in different categories of subjects. This increased muscle activation leads to a greater expression of explosive force during the push-up phase.

Studies in the scientific literature show that performing the SJ test with a starting position with knee angles less than 90 degrees allows for better vertical jumps [49,50]. These studies, however, were performed on male and female athletes, whereas the present study examined the effects of starting position on vertical jumping in female subjects only. A study on female subjects was necessary given the differences that exist between males and females, especially at the hip and knee joints [51,52]. This angle allows for a different accumulation of elasticity and potential energy in the muscles of the lower limbs and, in addition, may promote different force transmission and joint alignment during jumping [53,54]. A further study with the same conditions involving male non-athletes with similar ages and anthropometric characteristics would be needed to compare the results and identify any differences between genders. Finally, another important fact to consider is that the subjects were asked to jump upwards without being giving any instruction on what starting position to adopt. Overall, 62.5% of the subjects tested independently chose a knee starting position of about 70 degrees, and all of these subjects achieved jump heights greater than 16 cm, while the remainder, who chose knee starting angles below 70 degrees and above 80 degrees, achieved lower jump heights.

The data collected in our study pertain to jump values that are significantly lower than those found in the literature for elite athletes [55,56,57], who reach much higher jump heights (in our study the jump heights settled between 12 and 22 cm, while elite athletes can reach and exceed a jump height of 40 cm in the SJ test). This was precisely the discriminating element that led to us carrying out this study which aimed to identify the most suitable starting angles for subjects who, although in adequate psychophysical condition, are not athletes who are familiar with the gesture of the SJ and have trained to improve sport-specific sporting performance.

### 4.1. Limitations

The notable limitations of this study pertain to its relatively low sample size and the use of a non-probability sampling method. While the controlled laboratory environment and homogeneous sample were essential for isolating variables, the generalizability of the results to a broader population of students may be constrained. Future research with larger, more diverse samples could further validate and extend the current findings. Another limitation is that of having only considered the axis of rotation of the knee and not other equally important variables such as the position of the hip or spine. This limitation could be overcome in the future with the use of more sophisticated tracking systems that are able to highlight the relationships between the various segments of the body in the execution of the SJ. Further studies will need to focus on the jumping abilities of non-athlete male subjects with similar ages and physical characteristics, similar to the sample examined in this study, to clarify the differences between the two genders and apply this evidence in the training field.

### 4.2. Practical Applications

The findings of this study hold valuable implications for the training process, particularly in the field of improving physical performance in non-athlete female subjects, because they are the ideal subjects, having not yet made the greatest progress in increasing jump levels. Recognizing the importance of initiating knee angles, trainers and coaches might consider incorporating this knowledge into tailored training regimes to improve the physical efficiency of female individuals who do not have athletic performance goals. The optimization of one’s starting position, particularly at approximately 70 degrees knee angle, could be emphasized in training tools and methodologies to enhance functional and performance skills. Understanding the influence of starting positions on explosive jump elevation contributes to a more nuanced approach in training interventions. Based on the insights gained from this study, aspects such as muscle activation, joint alignment, and force transmission during jumping could be targeted more effectively.

## 5. Conclusions

The results of this study highlight the relationship between different initial knee flexion angles and elevation in explosive jumps performed by non-athletic female subjects partaking in the SJ test. This evidence is useful for physical and sporting activity professionals, as it could help them to improve their level of knowledge regarding the optimal execution of the gesture in a particular category of subjects and subsequently apply this knowledge within their professional activities by refining training strategy proposals and promoting more effective interventions to improve the physical efficiency of injured subjects. To improve functional and performance capabilities, the training process should meticulously address various variables. Based on the evidence presented, it is advisable to incorporate and optimize starting positions through the use of appropriate training tools and methodologies that differ based on numerous factors, including gender, exercise and sporting experience, and age.

## Figures and Tables

**Figure 1 jfmk-09-00026-f001:**
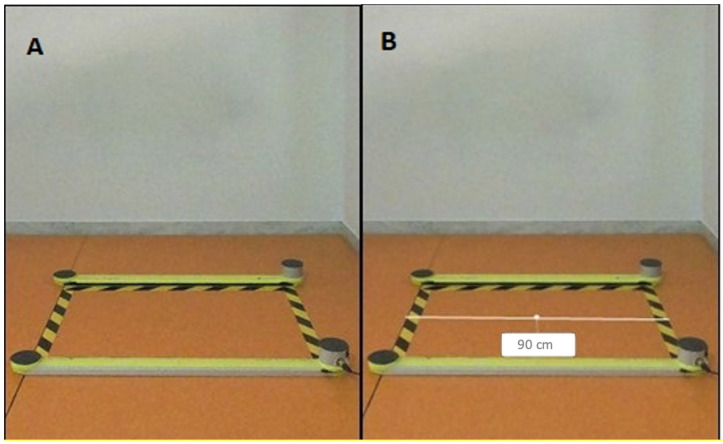
Calibration procedure with Kinovea software. In image (**A**), there is the frame of the acquisition area; in image (**B**), knowing the distance between the two lines, a corresponding value is given using Kinovea tools.

**Figure 2 jfmk-09-00026-f002:**
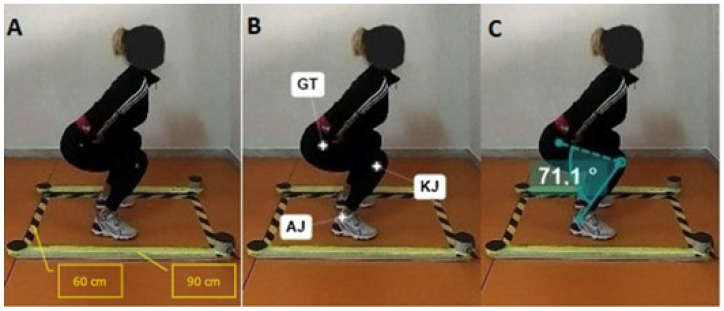
Image processing process with Kinovea. Image (**A**) shows the frame without processing; image (**B**) shows the identification of applied markers (GT = great trochanter; KJ = knee joint; AJ = ankle joint), and image (**C**) shows the starting knee angle.

**Figure 3 jfmk-09-00026-f003:**
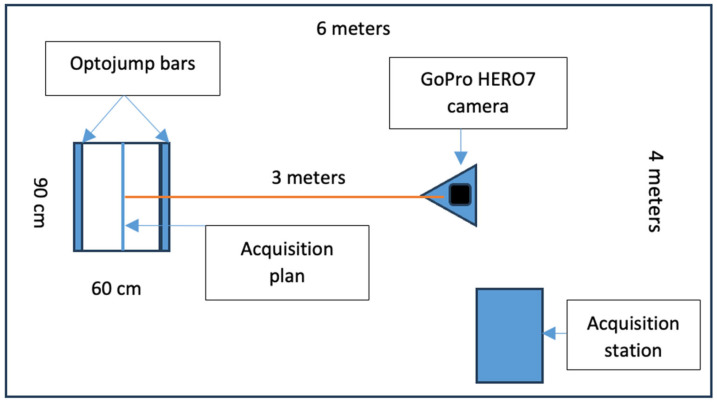
Laboratory setting with tools and measurements.

**Figure 4 jfmk-09-00026-f004:**
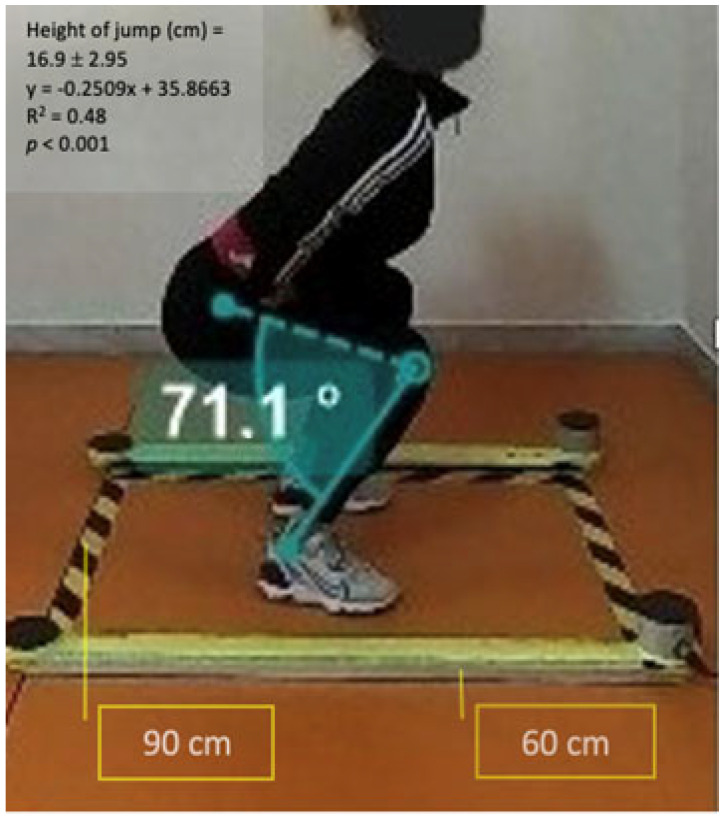
Example of knee flexion angle measurement in starting position with marking and tracking using Kinovea software.

**Figure 5 jfmk-09-00026-f005:**
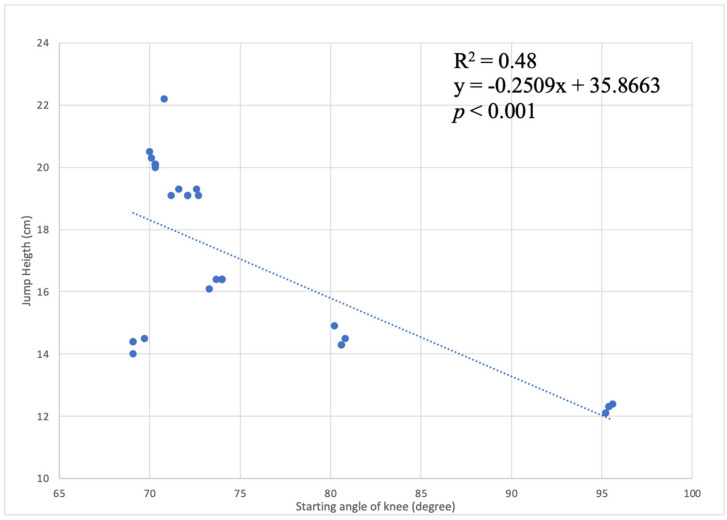
Correlation between knee starting angle and jump height attained by female athletes in the Squat Jump (SJ) test.

**Table 1 jfmk-09-00026-t001:** Biographical and anthropometric data of the study participants.

N = 32	Age (Years)	Weight (kg)	Height (cm)	BMI
Mean	23.37	58.68	166.96	21.05
Standard deviation	0.94	1.92	2.07	0.71

**Table 2 jfmk-09-00026-t002:** Measurements of knee starting angles and final jump height, along with correlation coefficients.

	Average	Correlation Coefficient	Adjusted R-Squared	Angle Coefficient	Sign. *p*
Height of jump (cm)	16.9 ± 2.95	−0.701	0.48	0.25	<0.001
Degree of knee flexion	75.578 ± 8.32				

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
