# Peer review of "Determining the Relationship between Squat Jump Performance and Knee Angle in Female University Students"

_jfmk, 2024, doi:10.3390/jfmk9010026_

Round 1
Reviewer 1 Report
Comments and Suggestions for Authors
Introduction:
Format: The text should be separated into paragraphs, where each one of them represents a line of argument of the introduction.
The argument used to justify the sample used is weak. The authors should improve this section.
Lines 53-54: More details are need related to results of references 18 and 19.
Lines 61-62: That sentence miss of more arguments, please amplify.
Lines 75-80: referes to a CMJ jump, try to modify or include arguments for SJ.
Objective: the wording of the objectives is not clear. It is recommended to rewrite to facilitate understanding of the main objective.
Likewise, the authors do not express any type of hypothesis regarding the main objectives of the study.
Methods:
Line 103: add units.
Line 103: Add more information about the anthropometric characteristics of the sample (BMI, weight, height, age, etc.)
Lines 105-107: What tool or classification did authors use to determine that they were not athletes?
Instruments
Lines 121-122: add some references for that instrument and for all instruments employed in the study. Also for all the software employed.
Add information about each instrument (Model, Brand, Country and city)
Expresions like: "itself played a crucial role in ensuring" (line 170); "was carefully prepared to accommodate the specific requirements of the study"; should be reduced in intensity.
Line 173: explain before the first use all the abbreviations.
In general, try to be more specific and concise in the description of the method.
Please add some kind of figure of the lab layout.
No markers were used in the degree assessment.
It would be useful to include a figure with the procedure to facilitate understanding.
More information about the protocol is needed. With current data it is not possible to reliably reproduce the study. It is unknown who performed the analysis, their experience, the error they may make by not using markers. The clothing conditions used by the study participants are unknown; according to Figure 1, they were worn with long pants, making it difficult to detect anatomical points.
The methodology used is poor and presents major biases that compromise the veracity of the results achieved.
The statistical analysis section is insufficient, and the evidence presented does not ensure the verification of the stated objectives.
The analyzed variables are not adequately described.
The figures are of low quality. Figure 2 lacks units and presents very little information.
When reading the summary of the work and the introduction I expected to find a controlled study of the knee angle used in a jumping battery (SJ), but when delving deeper into the document this was not the case.
The lack of rigor in the means and methods used in the present study makes its publication inadvisable.
Comments on the Quality of English LanguageModerate editing of English language required.
English should be reviewed by a native.
Reviewer 2 Report
Comments and Suggestions for Authors
Dear,
plese find my comments attached.
Kind regards

Reviewer 3 Report
Comments and Suggestions for Authors
CORRELATION BETWEEN KNEE ANGLE VARIATION AND SQUAT JUMP ELEVATION IN A SAMPLE OF FEMALE UNIVERSITY STUDENTS
General Commentary
This article presents a very interesting and pertinent question of research of influence knee angle of performance vertical jump.
However, some questions need to be clarified in order to better understand and apply the results found.
MAJOR CONSIDERATION
INTRODUCTION
Objective
I suggest that the authors aim to determine the joint angle based on the height of the jump, which would imply cause and effect, that is, to use linear or non-linear regression models in data analysis.
I don't see much point in using correlation models, which are not cause and effect. In this case, the correlation is just a complementary result of the regression.
MATERIALS AND METHODS
Study Design
I suggest that authors start the materials and methods with the subchapter "study design", where the type of study, briefly the entire design, and ethical precepts should be addressed. In addition, insert a figure with the study design in time line.
Participants
It is not clear in the study whether or not the participants have previous experience with performing this type of vertical jump (SJ).
Furthermore, it is also not clear why the authors chose the squat jump instead of a vertical jump that is easier to execute and apply, such as the countermovement jump?
GoPro 7 Hero
In the methods, the authors report that they used this GoPro camera, but do not comment on how the distances were corrected based on the distortion that this camera model presents due to its lens shape. Another possibility could be that the camera configuration allows this. I suggest that this is made very clear in the methods, otherwise the results will change.
Statistical Analysis
In addition to inserting a regression model, I suggest that authors classify the correlations performed (Null, Weak, Moderate, Strong, Very Strong...), use bibliographic references.
RESULTS
Figure 1.
It is not clear whether the authors used reflective markers on the hip, knee and ankle joints to carry out such an experiment. Could you clarify that? Furthermore, the quality of the figure is not good.
Table 2 and Figure 2.
The units of measurement for each variable are not shown. Regarding figure 2, it needs to be referred by adjusting scales (I suggest using the graph pad prism software for this).
DISCUSSION
Based on the comments cited above, the authors should review the entire discussion. Furthermore, it is not clear why only the range of movement of the knee would be related to the performance of a vertical squat jump, which isolates contractile elements (muscular strength) for its performance.
Could some reflections, and the influence of hip angles, affect such performance? What is the limitation of not having evaluated this?
In relation to range of movement, has it been established that it has no relation to the capacity to produce maximum force or rapid force? If so, wouldn't that explain the negative correlation? What other explanations for this?
Regarding the SJ vertical jump height with an average of 16.9 cm found? Amateur female athletes should have average SJ heels above 40, 45 cm to be considered good. In other words, can the low performance of SJ vertical jumps in the present study be considered a limitation?
Half of the discussion paragraphs do not present any literature references to discuss what was found, I suggest that this be reviewed by the authors.
Limitations and Practical Application
Please insert two final chapters (Limitations and Practical Application).
Limitations
Although some limitations are addressed by the authors, others discussed should be included in this subchapter.
Practical Application
What are the applications of the above in this study? Please describe in this chapter (perhaps the last figure should be included here)
CONCLUSION
The conclusion does not match the results found, please review all the writing, describing the results found in the study. Explanations of the findings should be addressed in the discussion.
Round 2
Reviewer 3 Report
Comments and Suggestions for Authors
Minor revisions
Title: "Correlation between knee angle variation and squat jump elevation in a sample of female university students"
As linear regression tests were added, I decided to remove the word correlations from the title of the manuscript and put "determine"
Suggestion: Determine squat jump performance with knee angle in female university students.
Regarding the figures, please add the units of measurement in all of them. And in the case of the linear equation, insert the equation and the r-square value in the image.
Author Response
Dear Reviewer,
Thank you for your valuable feedback on the title and figures of the manuscript. I appreciate your insights, and I have addressed each point below:
1. Title modification: Original title: "Correlation between knee angle variation and squat jump elevation in a sample of female university students". Revised Title: "Determine squat jump performance with knee angle in female university students."
We have considered your suggestion and modified the title accordingly. The revised title now emphasizes the aim of determining squat jump performance in relation to knee angle among female university students.
2. Inclusion of units in figures: You mentioned the need to add units of measurement to all figures. We have carefully reviewed the figures.
3. Linear equation and R-square value in figure: You suggested inserting the linear equation and the r-square value into the images containing the linear equation. We agree that this information is essential for a more comprehensive understanding of the results.
We appreciate your thorough review and constructive suggestions.
Best regards